# Pulmonary Embolism and Respiratory Deterioration in Chronic Cardiopulmonary Disease: A Narrative Review

**DOI:** 10.3390/diagnostics13010141

**Published:** 2023-01-01

**Authors:** Delphine Douillet, Tahar Chouihed, Laurent Bertoletti, Pierre-Marie Roy

**Affiliations:** 1Department of Emergency Medicine, University Hospital of Angers, Avenue of the Hotel Dieu, 49100 Angers, France; 2UMR MitoVasc CNRS 6215 INSERM 1083, University of Angers, 49100 Angers, France; 3FCRIN, INNOVTE, 42023 Saint-Étienne, France; 4Emergency Department, University Hospital of Nancy, Nancy-Université, 54000 Nancy, France; 5Department of Vascular and Therapeutic Medicine, University Hospital of Saint-Etienne, University Jean Monnet, Mines Saint-Étienne, INSERM, U 1059, CIC1408, 42023 Saint-Étienne, France

**Keywords:** pulmonary embolism, acute heart failure, trigger, chronic obstructive pulmonary disease, diagnostic strategy

## Abstract

Patients with chronic cardiopulmonary pathologies have an increased risk of developing venous thromboembolic events. The worsening of dyspnoea is a frequent occurrence and often leads patients to consult the emergency department. Pulmonary embolism can then be an exacerbation factor, a differential diagnosis or even a secondary diagnosis. The prevalence of pulmonary embolism in these patients is unknown, especially in cases of chronic heart failure. The challenge lies in needing to carry out a systematic or targeted diagnostic strategy for pulmonary embolism. The occurrence of a pulmonary embolism in patients with chronic cardiopulmonary disease clearly worsens their prognosis. In this narrative review, we study pulmonary embolism and chronic obstructive pulmonary disease, after which we turn to pulmonary embolism and chronic heart failure.

## 1. Introduction

Dyspnoea is one of the leading symptoms in patients presenting to emergency departments, and it has a wide range of possible causes. In a large prospective cohort, according to the physicians more than half of the cases of dyspnoea were attributed to respiratory pathology, one fifth to cardiac causes and less than 10% to mixed cardiac and respiratory causes [1]. In Europe and the Asia-Pacific region, the three main causes were lower respiratory tract infection (24.9%), heart failure (HF) (17.3%) and chronic obstructive pulmonary disease (COPD) exacerbation (15.8%) [2]. Pulmonary embolism (PE) is a rarer but potentially life-threatening cause of dyspnoea. New dyspnoea at rest or on exertion is the most frequent symptom in patients with PE [3,4]. 

Venous thromboembolism (VTE) is the consequence of the interaction between permanent patient-related risk factors and contextual transient risk factors. Congestive heart failure or respiratory failure are considered as moderate risk factors (odds ratio 2 to 9) [5,6]. It is important to note that they are both classified in the same group as active cancers and treated with estrogen-based hormone therapy. Indeed, many studies have assessed the incidence of VTE in hospitalized patients with pre-existing respiratory signs due to their chronic cardiopulmonary pathologies making the diagnosis of PE difficult. The incidence of PE, chronic heart failure and chronic respiratory failure also increases with age, and both conditions are among the most frequent causes of dyspnoea leading to hospitalization, relevant morbidity and mortality. COPD is the most common cause of chronic respiratory failure.

Despite important progress in the treatment of chronic heart failure or chronic respiratory failure, the management of acute exacerbation remains a challenge and is associated with an in-hospital mortality rate around 4% to 10% for acute HF (AHF) [7,8,9] and around 3% for COPD. One explanation could be the heterogeneity of symptom complexes, the difficulty to diagnose and the severity of precipitating factors. PE may be one important but underestimated factor. The diagnosis of PE in the context of the exacerbation of chronic heart or pulmonary disease remains a major issue. All the symptoms of exacerbations could be shared with the diagnosis of PE, possibly rendering PE a differential diagnosis. A French prospective survey that recruited consecutive patients presenting with suspected heart-failure-related dyspnoea was performed in 26 emergency departments [10]. Among the 158 patients in whom the diagnosis of HF was not retained, emergency physicians reported pulmonary disease in 51.3% of cases (mainly pulmonary infection, PE and COPD exacerbation), cardiac disease in 29.1% (mainly acute coronary syndrome), unspecific diagnosis (discomfort, stress, panting or palpitation) in 10.7%, renal failure in 5.1% and gastric disease in 3.8%.

Both underdiagnosis and overdiagnosis of PE are associated with higher morbidity and mortality rates. Untreated pulmonary embolism can be fatal [6,11], while overtreatment exposes the patient who does not have pulmonary embolism to an unjustified risk of major bleeding [6,12,13]. PE can therefore be a trigger, a differential diagnosis or occur secondarily in patients with acute exacerbation of HF or COPD.

While useful guidelines have been developed for diagnosing acute heart failure, COPD exacerbation and PE, no recommendation addresses the interrelatedness of these conditions, nor does searching for PE in the algorithm of these chronic disease decompensations shed any light [6,14,15]. The European Society of Cardiology (ESC) guidelines for acute heart failure note that “D-dimer levels should be measured when acute pulmonary embolism is suspected”, albeit without specifying which algorithms to use [14]. In the recommendations of the European Respiratory Society (ERS), the issue of the triggering factor is not addressed. There is no mention of venous thromboembolic disease [15]. In the French recommendations, the search for PE should be carried out in case of diagnostic uncertainty by a D-dimer test (to exclude a pulmonary embolism) [16]. On the contrary, a large, multicentre prospective cohort (116 emergency departments) confirmed that frequently the diagnostic management of PE does not adhere to guidelines. The risk for thromboembolism during follow-up was six times higher in these patients than in those who received appropriate management. Previously diagnosed heart failure or chronic lung disease are independent risk factors for inappropriate management with an odds ratio (OR) of 1.53 (95%CI 1.11 to 2.12) and OR: 1.39 (95%CI 1.00 to 1.94), respectively [17]. This area of overlap between chronic cardiopulmonary disease and venous thromboembolic disease is poorly understood and often overlooked in medical reasoning.

Indeed, there have been some major trials on COPD and PE, the results of which we will outline in this review, but very few studies exist on acute HF. In this narrative review, we will elaborate on the interrelationship between PE and COPD, as well as PE and heart failure, and describe the diagnostic approaches in these contexts of acute dyspnoea.

## 2. Pulmonary Embolism and Chronic Obstructive Pulmonary Disease

### 2.1. Physiopathology

The physiopathology of venous thrombosis is based on the association of venous stasis, damage to the venous endothelium and hypercoagulability. First of all, patients with COPD share other risk factors for VTE such as the association with usually sedentary lifestyle related to pulmonary hypertension and right-sided heart failure [18,19]. Furthermore, COPD is associated with a hypercoagulable state and coagulation dysfunction [20]. Many mechanisms, including systemic inflammation, eosinophilic inflammation, oxidative stress, endothelial dysfunction and hypoxemia may increase the risk of thromboembolism in COPD patients [21,22,23]. These contributing factors are enhances during the exacerbation of COPD, which exacerbates coagulation [24]. Chronic systemic inflammation in COPD is indicated by increased blood neutrophil count [25] and increased serum concentrations of inflammatory markers such as fibrinogen [26,27,28], C-reactive protein (CRP) [29,30], interleukin 6, interleukin 8 and TNFα [31,32]. Polycythaemia may also be a contributing factor promoting development of pulmonary hypertension and pulmonary endothelial dysfunction, but this has become rare due to the prescription of oxygen therapy in patients with COPD and hypoxemia [33].

### 2.2. Prevalence of PE in Patients with COPD Exacerbation

According to a post-mortem analysis, 20.9% of people who die from COPD had PE, and an incidence of 89.9% multiple thrombi in small pulmonary arteries and arterioles was found [34,35].

Although deep venous thrombosis (DVT) is two times more frequent than PE in the general population, COPD patients are more frequently diagnosed with PE than DVT [36]. Prospective small size cohort studies suggest a high prevalence of PE in acute exacerbation of COPD, ranging from 18 to 29% [37,38,39,40,41]. A few large general population studies have shown a modest excess risk of PE in COPD with odds ratios ranging from 2.51 (95%CI 1.62–3.87) to 5.46 (95%CI 4.25–7.02) [42,43].

In a meta-analysis, patients admitted to the hospital for an unexplained acute COPD exacerbation had a 16% prevalence of PE, with a wide 95%CI of 8.3% to 25.8% [44]. However, among the 7 studies included in this work, the definition of management (i.e., diagnostic algorithm), inclusion criteria, definition of COPD, clinical setting (outpatient vs. inpatients) and validation of the diagnosis of PE were not similar. All studies are prospective but with a small cohort size (the largest included 197 patients).

Couturaud et al. published the first high level of evidence study assessing the prevalence of PE among patients hospitalized with acute worsening respiratory symptoms [45]. It was a multicentre, cross-sectional study with prospective follow-up conducted in seven French hospitals. A predefined pulmonary embolism diagnostic algorithm based on Geneva score, D-dimer levels and spiral computed tomographic pulmonary angiography (CTPA) plus leg compression ultrasound was applied within 48 h of admission; all patients received follow-up at 3 months. Consecutive adult outpatients admitted for acutely worsening respiratory symptoms of COPD were eligible. A total of 740 patients were analysed. PE was diagnosed in 5.9% (95%CI 4.5–7.9%) and proximal deep vein thrombosis in 1.4% (95%CI 0.7–2.5%) of patients admitted to hospital with an acute worsening of respiratory symptoms. In this study, the prevalence of PE was higher in the subgroup of patients with “suspected PE” and close to what is observed in populations without pre-existing COPD in whom PE is suspected, i.e., around 10% (95%CI 7.1–14.0%). The most important message of this study was the significant prevalence of VTE, i.e., 3.2% (95%CI 1.9–5.3%), in COPD patients for whom the physicians did not suspect PE.

Shortly after Courturaud et al.’s study, the SLICE study was published [46]. Jimenez et al. performed the first randomized clinical trial to compare the standard of care with an active strategy for diagnosing PE. They excluded patients with an initial suspicion of PE. A total of 746 patients were randomized across 18 hospitals in Spain. In the interventional group, all patients had a D-dimer test and, where there was a positive result according to the laboratory of each centre, a CTPA was performed. A concomitant complete lower-limb compression ultrasound was performed. The overall rate of PE in the intervention group during the initial workup period was 4.6 (95%CI 2.7–7.3%). This study failed to demonstrate a difference between the two groups for the primary composite outcome (i.e., nonfatal new or recurrent symptomatic VTE, readmission for COPD or death at 90 days) possibly due to a lack of sensitivity. The primary outcome occurred for 29.7% in the intervention group (n = 110/746) and 29.2% in the control group (n = 107/764) (*p* = 0.86).

Two meta-analyses were then conducted assessing the VTE prevalence in patients admitted for acute exacerbation of COPD. The first one included 16 studies (N = 4093 patients and the estimated pooled prevalence of PE in patients was 12% (95%CI 9–16%; I^2^ = 94.8%) and the second included 17 studies in which the estimated pooled prevalence was 11% (95%CI 6.0–17.0%; I^2^ = 96.87%) [47,48]. However, the studies included were heterogeneous in terms of PE diagnostic work-up. Their groupings are questionable and provide little benefit compared to the PEP and SLICE studies, which were also pooled [49]. Among hospitalized patients with COPD exacerbation but no initial clinical suspicion of PE, the overall estimated pooled prevalence of PE was 3.6% (95% CI, 2.6–5.1%) (I^2^ = 0%; *p* = 1.0) [49].

### 2.3. Impact of PE Diagnosis for Patients with COPD Exacerbation

Patients with PE diagnosed in the context of COPD exacerbation have a worse prognosis than patients without PE [50]. In a post-hoc analysis of the PIOPED study, the 1-year mortality rate was 53.3% in the selected population of patients with COPD. This rate was twice as high as that of PIOPED patients in general and COPD patients without PE. After adjustment for patient characteristics, the estimated risk of dying within 1 year was 1.94 times higher (95%CI 1.17 to 3.24) for patients with COPD and pulmonary embolism compared with those without pulmonary embolism, and 1.14 (95%CI 0.85 to 1.54) times higher for patients without COPD (interaction *p* = 0.08) [51]. In a prospective cohort, Gunen et al. estimated the 1-year mortality rate as twice as high in the VTE subgroup of COPD patients hospitalized with an exacerbation (31.8 versus 61.9%) compared to patients without VTE. In addition to increased mortality, the length of hospital stay was 4–5 days longer in the patients with VTE (*p* < 0.001) [41]. In the PEP trial, the proportion of patients who died during follow-up was higher among those with VTE at admission versus patients without VTE (risk difference 20.7% (95%CI 10.7–33.8%; *p* < 0.001) [45]. However, in the VTE group, the mortality was mainly due to cancer, thus highlighting the poorer prognosis when associated to VTE and COPD. While in patients with COPD who had an acute episode of VTE, the risk of recurrent VTE was not any higher than that in non-COPD patients [52]. COPD was not associated with an increased risk of VTE recurrence on univariate or multivariate analyses (hazard ratio: 1.0 (95%CI 0.7–1.4)).

It is important to consider that in this population where smoking is the cause of COPD, the diagnosis of venous thromboembolism should lead to a systematic search for cancer.

### 2.4. Diagnostic Strategy/Systematic Screening for VTE

In the PEP study, VTE events occurred in 3.2% of patients for whom PE was not suspected. The issue now is whether this rate justifies a systematic search for pulmonary embolism or whether this risk remains acceptable. This risk is not sustainable as the combination of a COPD exacerbation and PE will significantly worsen the patient’s prognosis. The treatment of both components, therefore, seems to be essential. Systematic screening exposes an increase in the number of CTPAs. This has already been the case for 20 years without improving patient outcomes [53,54]. Some authors suggest that there is a need to assess the utility of D-dimer levels in high-risk predictions [55].

Another major issue is the reliability of a different score aiming to assess the initial pre-test probability. The presence of subjective items such as “PE is the most likely diagnosis” makes the assessment biased by the presence of COPD. The Wells score, the YEARS approach and the PeGED approach are therefore possibly influenced by these two interrelated conditions [56,57,58,59]. The most objective approach would be to use the revised Geneva score and then apply an age-adjusted threshold [60,61]. However, heart rate can be largely influenced using B2 agonist, which is a mainstay of treatment for COPD exacerbations.

The other approach would be to use the gestalt. Gestalt has sometimes been shown to be superior to the pre-test evaluation scores [62]. When evaluating a patient with acute exacerbation of COPD for PE, clinical gestalt may be even more reliable in determining the risk of PE because the Wells criteria, the revised Geneva criteria, and the criteria for ruling out PE do not take a history of COPD into account when determining PE risk.

Future studies should determine the most effective, safe and cost-effective strategy for patients with acute exacerbation of COPD.

The diagnostic performance of the D-dimer is moderately influenced by the presence of COPD [63]. The yield of D-dimer is 0.5 to exclude PE according to the SLICE study [46]. However, this study used fixed, non-age-adjusted thresholds. A post-hoc analysis with adjusted D-dimer resulted in a 14% absolute increase of negative D-dimer.

Performing a CTPA made it possible to identify alternative diagnosis in a large proportion of patients and more than 20% had therapeutic consequences and specific treatment [46].

Given these discrepancies, the diagnostic value of clinical scoring systems and the best strategy for excluding PE are ripe for further investigation for patients with COPD exacerbation.

## 3. Pulmonary Embolism and Acute Heart Failure

### 3.1. Physiopathology

Firstly, it is important to consider that established cardiovascular risk factors, including hypertension, obesity, dyslipidaemia, hypertension, diabetes, metabolic syndrome, estrogen therapy and smoking, also increase the risk of VTE [6]. Metabolic syndrome has been associated with a higher risk of VTE [64,65,66]. A registry of 20,374 patients demonstrated that metabolic syndrome increased the risk with a hazard ratio (HR) of 1.84 (95%CI 1.30–2.59) [65]. Furthermore, venous thromboembolism and heart failure are strongly age-related.

In addition to these risk factors, endothelial injuries, inflammation and systemic and local hypercoagulability play an integral mechanistic role in the pathophysiology of atherothrombosis and contribute to venous thromboembolic events. Endothelial damage plays a pathophysiological role in the development of VTE. Markers of inflammation such as C-reactive protein and fibrinogen have been found in chronic heart failure [67]. Statins have been shown to be effective in reducing venous thromboembolic risk in a randomized controlled trial in primary prevention reducing the rate of symptomatic VTE by 43% in patients with elevated C-reactive protein and LDL cholesterol levels <130 mg/dL compared with placebo [68]. All these factors may promote the occurrence of both atherothrombosis (leading to myocardial infarction and HF) and venous thrombosis (leading to PE).

Conversely, PE can lead to a rapid rise in pulmonary pressures and acutely precipitate HF by causing right ventricular dysfunction (acute cor pulmonale) [69]. More than 50% of the pulmonary vasculature must be obstructed by thrombosis for an increase in pulmonary pressure to occur [70]. This can lead to arterial hypotension and impaired oxygenation, which can facilitate the development of myocardial ischemia and of arrhythmias such as atrial fibrillation. This further degrades cardiac function and can lead to profound haemodynamic instability and shock.

### 3.2. Prevalence of PE in Patients with Acute Heart Failure

Most of the data on the association between and prevalence of acute heart failure and PE comes from post-mortem studies. A large Swedish study with 23,796 autopsies performed in the general population found that the frequency of PE increased according to the presence of intracardiac thrombi, which were present in 1706 (7.2%) patients [71]. PE prevalence in patients with isolated left thrombi, isolated right thrombi and combined thrombi was 28.5%, 35.6% and 48.9%, respectively. In a study of 1032 autopsies in patients with heart disease, Pulido et al. found 231 cases (24.4%) of PE; 100 of these patients were diagnosed with massive PE [72]. A higher rate was found by Saad et al. with 34.1% of patients with PE diagnosed post-mortem in patients who had died in a hospital specializing in cardiology [73]. The most common diagnostic discrepancy between the initial clinical and post-mortem diagnoses was PE. Following this, De Macedo et al. reviewed 1457 autopsies and analysed 595 patients with heart failure [74]. All patients’ autopsy reports were reviewed to assess the occurrence of thromboembolic events. The main cause of death was progressive HF in 253 (42.5%) patients, infections in 112 (18.8%), myocardial infarction in 86 (14.5%) and pulmonary embolism in 81 (13.6%). The most frequent thromboembolism was pulmonary embolism in 135 (36.1%) patients; in 81 events (60%), it was considered the cause of death. This evidence confirms the frequent presence of PEs among the death causes of patients hospitalized for heart failure.

In a case-control study, Howell et al. showed that chronic HF is an independent risk factor for venous thromboembolism, and the risk increases markedly with decreasing left ventricular ejection fraction. Indeed, in patients with an ejection fraction of less than 20%, the OR of venous thromboembolism was of 38.3 (95%CI 9.6, 152.5) [75].

A nationwide, population-based case-control study performed in Denmark found that heart disease increases the risk of pulmonary embolism [76]. Indeed, myocardial infarction and heart failure in the preceding 3 months conferred high risks of PE with an OR of 43.5 (95%CI 39.6–47.8) and 32.4 (95%CI 29.8–35.2), respectively. Right-sided valvular disease was associated with a higher risk (OR 74.6 [95%CI 28.4–195.8]) than left-sided valvular disease (OR 13.5 [95%CI 11.3–16.1]). The risk estimates were substantially more pronounced when venous thromboembolism was the second diagnosis mentioned in the record, indicating that venous thromboembolism was a complication of the heart disease and not vice versa.

In a large retrospective study using data from the National Hospital Discharge Survey, among 181,000 hospitalized patients with chronic HF, PE was diagnosed in 0.73%, DVT in 1.03%, and VTE in 1.63% [77] of cases. But this was a study with many biases and its retrospective nature makes it impossible to assess the real prevalence.

In 2005, Darze et al. performed the first prospective study on this population aiming to assess the prevalence of PE during hospitalization [78]. A total of 1417 patients were admitted to the coronary care unit during the study period. Of 198 patients with severe chronic heart failure recruited in this study, 18 patients (9.1%) received a diagnosis of PE during their hospitalization period. However, investigation for PE was initiated at the discretion of the attending physician. Only 36 patients were investigated, and half had confirmation of the diagnosis of PE. In 67% of the patients with PE, the diagnosis was confirmed within 5 days of hospital admission. None of these patients received an initial diagnostic work-up.

To the best of our knowledge, there is no study with a good level of evidence to date to determine the true prevalence of VTE events on admission of a patient for acute HF or during the following months.

### 3.3. Impact of PE Diagnosis for Patients with Acute Heart Failure

In a prospective cohort of patients admitted to coronary care with severe decompensated HF, 198 were included and 9.1% had a PE [79]. In a multiple logistic regression analysis, PE remained an independent predictor of death or rehospitalization at 3 months (OR: 4.0 [95%CI 1.1–15.1; *p* = 0.038]). The ESC-HF prospective, multicentre, observational survey conducted in 136 cardiology centres across 12 European countries assessed the outcomes of patients with HF. In this study, the rate of death related to PE was 5.6% in patients admitted for acute HF [80]. Moreover, in patients with PE, history of HF has been consistently associated with a higher mortality rate. In a large European registry (RIETE registry), Laporte et al. assessed clinical predictors for fatal PE in 15,520 patients with VTE [11]. The risk was 2 to 3 times higher in patients with cardiac or respiratory disease. Furthermore, among non-anticoagulated patients, inappropriateness of diagnostic management was associated with an increase of venous thromboembolic events or unexplained sudden deaths during the 3-month follow-up (absolute risk difference: 6.5%) [17].

### 3.4. Diagnostic Strategy/Systematic Screening for VTE

To date, no study has evaluated systematic screening to determine the prevalence of PE. The search for PE should be based on a pre-test assessment of the suspicion of PE. However, as with COPD patients, pre-test assessment scores were not initially performed for these patients with chronic HF. The Wells score and the derived YEARS and PeGED strategies also suffer from the question “is PE the most likely diagnosis?” restricting their usability. The revised Geneva score might be more relevant in this context, but this needs to be confirmed. All scores take into account the heart rate. In chronic heart failure, patients are often on anti-arrhythmic drugs such as betablockers, which distorts the assessment. Tachycardia is also possible with or without an arrythmia in the context of acute HF. However, no studies have evaluated the different diagnostic strategies in these patients with HF. D-dimer tests are often elevated (false positive) in this patient population [6,81]. However, they appear to retain good performance in excluding the diagnosis of PE. The age-adjusted threshold has not been tested in this specific population [61].

If no side effects of the CTPA were mentioned in the recent study of de Martini et al. assessing the feasibility and accuracy of diagnosing acute HF with CTPA in emergency department patients [82], diagnostic support should initially and firstly be performed by Point-of-Care Ultrasound (POCUS). Firstly, the use of compression ultrasonography in the diagnosis of DVT allows the diagnosis of PE to be made in the presence of respiratory signs and treatment to be started [6]. Secondly, an echocardiography should be performed to look for differential diagnosis in case of shock (hypovolemia, aortic dissection, pericardial tamponade, acute valvular dysfunction). A right ventricular to left ventricular ratio greater than 1 indicates the severity of PE. The Tricuspid Annular Plane Systolic Excursion (TAPSE) measures the longitudinal contraction of the right ventricle. It is taken in 4-cavity slice, in TM mode, targeting the tricuspid annulus in its lateral portion. It is considered that a TAPSE lower than 16mm is indicative of a right ventricular dysfunction [6,83]. The results of this echocardiography will drive therapeutic strategy for severe PE. Indeed, the physician will be able to introduce an anti-thrombotic strategy adapted to the patient’s clinical condition (anticoagulant or thrombolytic) and postpone the CTPA for 48 h after the control of the acute HF.

## 4. Conclusions

Pathologies with cardiorespiratory signs are sometimes intertwined in the acute phase, making global management difficult. However, it is through clinicians considering each of the components that the patient’s prognosis will improve. Physicians are therefore torn between conducting a systematic search that is costly in terms of time and resources and often useless on the one hand, and the risk of missing out on overlapping pathologies that can worsen the prognosis on the other. In patients with chronic pulmonary disease and patients with chronic heart failure admitted to hospital with worsening respiratory conditions, the prevalence of pulmonary embolism is not negligible. However, the data on heart failure patients are very weak, so larger studies are required. Further research is needed to understand the potential impact of routine screening for pulmonary embolism in these patient populations and in acute heart failure in particular.

## Data Availability

No new data was created or analysed in this study. Data sharing is not applicable to this article.

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
