# Peer review of "Pulmonary Embolism and Respiratory Deterioration in Chronic Cardiopulmonary Disease: A Narrative Review"

_diagnostics, 2023, doi:10.3390/diagnostics13010141_

Round 1
Reviewer 1 Report
Douillet et al. performed a well-documented narrative review of the complexity of pulmonary embolism (PE) diagnosis management and its consequences during two frequent cardiopulmonary acute conditions: chronic obstructive pulmonary disease (COPD) exacerbation and acute heart failure (AHF). Indeed, dyspnea is the cardinal symptom of all three conditions but PE is the less frequent cause of dyspnea in the ER, while COPD and AHF are more frequently involved. However, both COPD and AHF are recognized transient risk factors for VTE/PE and are associated with a worse prognosis when combined to PE.
As such, for each of these two comorbid associations (COPD – PE and AHF-PE), authors beautifully:
(1) depicted their physiopathological connections based on both coagulation abnormalities and endothelial dysfunction triggering their co-occurrence
(2) detailed the relevant literature data on PE prevalence in COPD/AHF
(3) highlighted the impact of PE diagnosis on the prognosis of COPD/AHF
(4) discussed the appropriateness of general PE diagnostic algorithms and their applicability to these two conditions
The perspective taken by the authors in the conception of this narrative review is highly original and offers a new vision of the importance of an appropriate PE diagnosis in cardiopulmonary pathologies that are commonly referred as differential diagnosis of PE as they share similar symptoms.
As pointed out by the authors, the lack of specific PE diagnostic algorithms in these two very frequent disorders supports the need to elaborate specific prediction rules.
Overall, the article is well structured and offers a recent and exhaustive documentation on the subject.
Here are some suggestions for text modification (i.e. some wording suggestions):
Line 36: “risk factors and contextual, usually temporary, risk factors” à transient
Lines 39: “It is important to note that they are both classified in the same group as active cancers and treated with estrogenic hormone” à estrogen-based hormone therapy
Line 79: Diagnosed heart failure or chronic lung disease are independent risk factors for inappropriate management à Previously diagnosed/previously known
Line 92: First of all, patients with COPD share other risk factors for VTE such as the association with usually sedentary pulmonary hypertension and right-sided heart failure à sedentary lifestyle related to pulmonary hypertension…
Line 98: These contributing factors are accentuated during the exacerbation of COPD, which exacerbates coagulation à enhances
Line 98: Prolonged systemic inflammation in COPD à Persistent/chronic
Line 111 Past retrospective or small studies suggest a high prevalence of PE in acute exacerbation àProspective small size cohort studies… (NB: ref 39 is a systematic literature review and all the other references are prospective studies)
Line 131: In this study, the prevalence of PE was close to what is observed in populations without pre-existing COPD in whom PE is suspected, i.e., around 10% (95%CI 7.1-14.0%). à In this study, the prevalence of PE was higher in the subgroup of patients with “suspected PE” and close to what is observed in populations without pre-existing COPD in whom PE is suspected, i.e., around 10% (95%CI 7.1-14.0%)
Line 139: A total of 746 patients were randomized across 18 hospitals in Spain. In the interventional group, all patients had a D-dimer test and, where there was a positive result according to the laboratory of each centre, a CTPA. à a CTPA was performed/obtained.
Line 148: Two meta-analyses were then made assessing the VTE prevalence in patients àconducted
Line 152: However, the studies included were heterogeneous in terms of finding the diagnosis of PE.à PE diagnostic work-up
Line 186: Systematic screening exposes an increase in the number of CTPAs.à exposes to an increase
Line 259: This evidence confirms the frequent presence of PEs in the death of patients hospitalized for heart failure à among the death causes
Line 263: Indeed, in patients with ejection fraction, under 20% of the OR of venous thromboembolism was of 38.3 (95%CI 9.6, 152.5). à an ejection fraction of less than 20%, the OR…
Line 291: In a prospective cohort of patients admitted to coronary care with severe decompensated HF à coronary care unit
Line 300: Furthermore, among non-anticoagulated patients, inappropriateness was associated with an increase of venous thromboembolic events or unexplained sudden deaths during 3-month follow-up (absolute risk difference: 6.5%)à inappropriateness of diagnostic management
Line 318: If, no side effects of the CTPA was mentioned in the recent study of de Martini et al àwere
Line 321: the use of compression ultrasonography in the diagnosis of DVT allow the diagnosis of PEà allows
Line 340: In patients with chronic obstructive bronchitisà pulmonary disease

Author Response
Editorial Board Members
Diagnostics Journal
December 28, 2022
Dear Editor-in-Chief,
We thank you for your attention to our study entitled " Pulmonary embolism and respiratory worsening of chronic cardiopulmonary disease: a narrative review.” and for giving us the opportunity to improve our manuscript.
Please find enclosed a point-to-point answer to the very useful comments from the Reviewers.
We think that the Reviewers suggestions and the associated revisions have substantially improved the quality of the manuscript. We hope that the editorial staff and Reviewers will find this revised version suitable for publication in the “Diagnostics Journal”.
Yours sincerely,
On behalf of all the authors,
Dr. Delphine Douillet and Prof. Pierre-Marie Roy
Letter Revised Manuscript
Please find below our point-by-point responses to all issues raised by the reviewers and the changes we made in our manuscript.
Reviewer #1:
Douillet et al. performed a well-documented narrative review of the complexity of pulmonary embolism (PE) diagnosis management and its consequences during two frequent cardiopulmonary acute conditions: chronic obstructive pulmonary disease (COPD) exacerbation and acute heart failure (AHF). Indeed, dyspnea is the cardinal symptom of all three conditions but PE is the less frequent cause of dyspnea in the ER, while COPD and AHF are more frequently involved. However, both COPD and AHF are recognized transient risk factors for VTE/PE and are associated with a worse prognosis when combined to PE.
As such, for each of these two comorbid associations (COPD – PE and AHF-PE), authors beautifully:
(1) depicted their physiopathological connections based on both coagulation abnormalities and endothelial dysfunction triggering their co-occurrence
(2) detailed the relevant literature data on PE prevalence in COPD/AHF
(3) highlighted the impact of PE diagnosis on the prognosis of COPD/AHF
(4) discussed the appropriateness of general PE diagnostic algorithms and their applicability to these two conditions
The perspective taken by the authors in the conception of this narrative review is highly original and offers a new vision of the importance of an appropriate PE diagnosis in cardiopulmonary pathologies that are commonly referred as differential diagnosis of PE as they share similar symptoms.
As pointed out by the authors, the lack of specific PE diagnostic algorithms in these two very frequent disorders supports the need to elaborate specific prediction rules.
Overall, the article is well structured and offers a recent and exhaustive documentation on the subject.
Thank you very much for this comment.
Here are some suggestions for text modification (i.e. some wording suggestions):
Line 36: “risk factors and contextual, usually temporary, risk factors” à transient
We agree with this text modification.
Lines 39: “It is important to note that they are both classified in the same group as active cancers and treated with estrogenic hormone” à estrogen-based hormone therapy
We agree with this text modification.
Line 79: Diagnosed heart failure or chronic lung disease are independent risk factors for inappropriate management à Previously diagnosed/previously known
We agree with this text modification.
Line 92: First of all, patients with COPD share other risk factors for VTE such as the association with usually sedentary pulmonary hypertension and right-sided heart failure à sedentary lifestyle related to pulmonary hypertension…
We agree with this text modification.
Line 98: These contributing factors are accentuated during the exacerbation of COPD, which exacerbates coagulation à enhances
We agree with this text modification.
Line 98: Prolonged systemic inflammation in COPD à Persistent/chronic
We agree with this text modification.
Line 111 Past retrospective or small studies suggest a high prevalence of PE in acute exacerbation àProspective small size cohort studies… (NB: ref 39 is a systematic literature review and all the other references are prospective studies)
We agree with this text modification.
Line 131: In this study, the prevalence of PE was close to what is observed in populations without pre-existing COPD in whom PE is suspected, i.e., around 10% (95%CI 7.1-14.0%). à In this study, the prevalence of PE was higher in the subgroup of patients with “suspected PE” and close to what is observed in populations without pre-existing COPD in whom PE is suspected, i.e., around 10% (95%CI 7.1-14.0%)
We agree with this text modification.
Line 139: A total of 746 patients were randomized across 18 hospitals in Spain. In the interventional group, all patients had a D-dimer test and, where there was a positive result according to the laboratory of each centre, a CTPA. à a CTPA was performed/obtained.
We agree with this text modification.
Line 148: Two meta-analyses were then made assessing the VTE prevalence in patients àconducted
We agree with this text modification.
Line 152: However, the studies included were heterogeneous in terms of finding the diagnosis of PE.à PE diagnostic work-up
We agree with this text modification.
Line 186: Systematic screening exposes an increase in the number of CTPAs.à exposes to an increase
We agree with this text modification.
Line 259: This evidence confirms the frequent presence of PEs in the death of patients hospitalized for heart failure à among the death causes
We agree with this text modification.
Line 263: Indeed, in patients with ejection fraction, under 20% of the OR of venous thromboembolism was of 38.3 (95%CI 9.6, 152.5). à an ejection fraction of less than 20%, the OR…
We agree with this text modification.
Line 291: In a prospective cohort of patients admitted to coronary care with severe decompensated HF à coronary care unit
We agree with this text modification.
Line 300: Furthermore, among non-anticoagulated patients, inappropriateness was associated with an increase of venous thromboembolic events or unexplained sudden deaths during 3-month follow-up (absolute risk difference: 6.5%)à inappropriateness of diagnostic management
We agree with this text modification.
Line 318: If, no side effects of the CTPA was mentioned in the recent study of de Martini et al àwere
We agree with this text modification.
Line 321: the use of compression ultrasonography in the diagnosis of DVT allow the diagnosis of PEà allows
We agree with this text modification.
Line 340: In patients with chronic obstructive bronchitisà pulmonary disease
We agree with this text modification.
Reviewer #2:
Dear Authors! I would like to congratulate you for the well work done. The review discusses a very important problem that leads to a lot of difficulties in a real practice. The pulmonary embolism diagnostics in a patient with a pre-existed cardiopulmonary disease is troublesome.
The only remark that I have is that you claimed to discuss the situation in which pulmonary embolism is suspected in a patient with chronic cardiopulmonary disease. At the same time, you then discuss diagnostic problems in patients with acute heart failure. I recommend you to correct the title of the paper to make it reflecting the content of the paper more correctly.
Thank you for your comments. We discuss the place of PE in decompensations of chronic pathologies mainly: COPD exacerbation and acute heart failure. It is in these situations that it is difficult to make the diagnosis of pulmonary embolism. We have changed the title: " Pulmonary embolism and respiratory worsening of chronic cardiopulmonary disease: a narrative review.”
Reviewer 2 Report
Dear Authors! I would like to congratulate you for the well work done. The review discusses a very important problem that leads to a lot of difficulties in a real practice. The pulmonary embolism diagnostics in a patient with a pre-existed cardiopulmonary disease is troublesome.
The only remark that I have is that you claimed to discuss the situation in which pulmonary embolism is suspected in a patient with chronic cardiopulmonary disease. At the same time, you then discuss diagnostic problems in patients with acute heart failure. I recommend you to correct the title of the paper to make it reflecting the content of the paper more correctly.
Author Response

(The authors gave the same response as above.)
